

# Tracking down global $NH_3$ point sources with wind-adjusted superresolution

Lieven Clarisse[1], Martin Van Damme[1], Cathy Clerbaux[2,1], and Pierre-François Coheur[1]

[1]Université libre de Bruxelles (ULB), Atmospheric Spectroscopy, Service de Chimie Quantique et Photophysique, Brussels, Belgium
[2]LATMOS/IPSL, Sorbonne Université, UVSQ, CNRS, Paris, France

**Correspondence:** Lieven Clarisse (lclariss@ulb.ac.be)

**Abstract.** As a precursor of atmospheric aerosols, ammonia ($NH_3$) is one the primary gaseous air pollutants. Given its short atmospheric lifetime, ambient $NH_3$ concentrations are dominated by local sources. In a recent study, Van Damme et al. (2018) have highlighted the importance of $NH_3$ point sources, especially those associated with feedlots and industrial ammonia production. Their emissions were shown to be largely underestimated in bottom-up emission inventories. The discovery was made

possible thanks to the use of oversampling techniques applied on 9 years of global daily IASI $NH_3$ satellite measurements. Oversampling allows to increase the spatial resolution of averaged satellite data, beyond what the satellites natively offer. Here, we apply for the first time the so-called superresolution techniques, which are commonplace in many fields that rely on imaging, to measurements of an atmospheric sounder, whose images consist of just single pixels. We demonstrate the principle on synthetic data and on IASI measurements of a surface parameter. Superresolution is a priori less suitable to be applied on

measurements of variable atmospheric constituents, in particular those affected by transport. However, by first applying the so-called wind-rotation technique, which was introduced in the study of other primary pollutants, superresolution becomes highly effective to map $NH_3$ at very high spatial resolution. We show in particular that it allows revealing plume transport in much greater detail than what was previously thought to be possible. Next, using this wind-adjusted superresolution technique, we introduce a new type of $NH_3$ map that allows to track down point sources much more easily than the regular oversampled

average. On a subset of known emitters, it allows to locate the source within a median distance of 1.5 km. We subsequently present a new global point source catalog consisting of more than 500 localized and categorized point sources. Compared to our previous catalog, the number of identified sources more than doubled. In addition, we refined the classification of industries into five categories: fertilizer, coking, soda ash, geothermal and explosive industry; and introduced a new urban category for isolated $NH_3$ hotspots over cities. The latter mainly consists of African megacities, as clear isolation of such urban hotspots is

almost never possible elsewhere due to the presence of a larger diffuse background. The techniques presented in this paper can most likely be exploited in the study of point sources of other short-lived atmospheric pollutants such as $SO_2$ and $NO_2$.





# 1   Introduction

As one of the primary forms of reactive nitrogen, $NH_3$ is essential in many of the Earth's biogeochemical processes. It is naturally present along with the nitrogen oxides in the global nitrogen cycle (Canfield et al., 2010; Fowler et al., 2013). However, the discovery of ammonia synthesis through the Haber-Bosch process in the early 1900s has made this vital compound

available in almost unlimited quantities, supporting the explosive population growth in the last century (Erisman et al., 2008). As a result, the nitrogen cycle is currently perturbed beyond the safe operating space for humanity, which has led to a host of environmental and societal problems (Steffen et al., 2015). The most obvious direct impact of excess $NH_3$ is that on air quality, as atmospheric $NH_3$ is one of the main precursors of secondary particulate matter, which has important adverse health impacts (Lelieveld et al., 2015; Bauer et al., 2016). Emissions of the two other important precursors ($SO_2$ and $NO_x$) have thanks to

effective legislation drastically decreased in the past twenty years in Europe and North America, and have started to level off in East Asia (Aas et al., 2019; Georgoulias et al., 2018). In contrast, no such decreases are observed or expected in the near future for $NH_3$ (e.g. Warner et al. (2017); Sutton et al. (2013)). Unlike the other precursors, $NH_3$ emissions are not well regulated, and in fact, the focus on decreasing $NO_x$ and $SO_2$ has already shown adverse effects on $NH_3$ emissions (Chang et al., 2016) and concentrations (Lachatre et al., 2018; Liu et al., 2018).

The lack of a global regulative framework stems in part from the historical relative difficulty in measuring $NH_3$ concentrations. Satellite-based measurements of $NH_3$, which were discovered about a decade ago, offer an attractive complementary means of monitoring $NH_3$. Satellite datasets have now reached sufficient maturity to be directly exploitable, even when the individual measurements come with large and variable uncertainties. Using satellite observations we have recently shown the importance of ammonia points sources on regional scales (Van Damme et al., 2018). In total over 240 of the world's strongest

point sources were identified, categorized and quantified. Somewhat expectedly, many of these point sources (or clusters thereof) were found to be associated with so-called 'concentrated animal feeding operations' (CAFOs) (Zhu et al., 2015; Yuan et al., 2017). However, much more surprisingly was the number of industrial emitters that was found, and in particular those associated with ammonia and urea-based fertilizer production. An evaluation of the EDGAR inventory in addition showed that emission inventories vastly underestimate the majority of all point source emissions, even when a conservative average $NH_3$

lifetime is assumed in the calculation of the satellite derived fluxes. Industrial processes could therefore be extremely important, especially on a regional scale. Altogether, these findings were made possible due to the availability of the large multiyear $NH_3$ dataset (Whitburn et al., 2016; Van Damme et al., 2017) derived from measurements of the IASI spaceborne instrument (Clerbaux et al., 2009), and the oversampling technique that was applied to sufficiently resolve localised emitters.

Oversampling techniques applied on measurements of satellite sounders allow to obtain average distributions of atmospheric

constituents at a higher spatial resolution than the original measurements (Sun et al., 2018). They exploit the fact that the footprint on ground of satellite measurements varies in location, size and shape each time the satellite samples an area. When pixels partially overlap, some information becomes available on their (sub-pixel) intersection. A high resolution mapping can however only be obtained by combining typically many hundreds of measurements. A crucial condition on which oversampling relies, is that the pixel centre and ground instantaneous field of view (GIFOV) of satellite measurements is known with a high





accuracy (typically $<1$ km, as opposed to the coarse spatial resolution of the extent of the satellite pixel which is typically $>10$ km). Practical implementation of oversampling is relatively straightforward once the footprint is known: a fine subgrid is constructed where the value of each cell of the grid is obtained as the average value of all overlapping GIFOVs. Optionally, the averaging can be weighted, to take into account measurement error, total pixel surface area and spatial response function. We

refer to Sun et al. (2018) and Van Damme et al. (2018) for comprehensive reference material on averaging and oversampling, detailed algorithmic descriptions and practical considerations for their implementation.

Oversampling has gradually become commonplace in the field of atmospheric remote sensing, especially in the study of short-lived pollutants such as $NO_2$ (Wenig et al., 2008), $SO_2$ (Fioletov et al., 2011, 2013), HCHO (Zhu et al., 2014) and $NH_3$ (Van Damme et al., 2014, 2018). The increased spatial resolution allows in first instance a much better identification of

emission (point) sources, quantification of their emissions (Streets et al., 2013) and study of transport and plume chemistry (de Foy et al., 2009). Oversampling applied to the study of point sources becomes even more useful when wind fields are taken into account. Beirle et al. (2011) showed that binned averaging per wind direction allows simultaneous estimates of both emission strengths and atmospheric residence times. Valin et al. (2013) and Pommier et al. (2013) introduced the wind-rotation technique, whereby each observation is rotated around the presumed point source according to the horizontal wind direction,

effectively yielding a distribution where the winds blow in the same direction. As we will also demonstrate, this reduces the overall spread of the transported pollutants and reduces contributions of nearby sources. Combining plume rotation with oversampling has proven to be a very successful technique for the study of $NO_2$ and $SO_2$ point sources, leading to massively improved inventories and emission estimates, and better constraints on the atmospheric lifetime of these pollutants (Fioletov et al., 2015; Wang et al., 2015; de Foy et al., 2015; Lu et al., 2015; Liu et al., 2016; McLinden et al., 2016; Fioletov et al., 2016,

2017).

However, as pointed out in Sun et al. (2018), while oversampling offers an increased resolution, it still yields a smoothed representation of the true distributions. There exists a large field of research, collectively referred to as superresolution (Milanfar, 2010) that attempts to construct high resolution images from several, possibly moving or distorted, low resolution representations of the same reality. Oversampling is in essence the simplest way of performing superresolution, but in a way that does

not fully exploit the spatial information of the measurements. Superresolution has been applied before in the field of remote sensing of land(cover) (e.g. Boucher et al. (2008); Xu et al. (2017)), but even though theoretically possible, it has not been applied to atmospheric sounding measurements. In this case, the 'images' as taken by sounders, are of the lowest resolution, i.e. they correspond to single, uniformly colored pixels. Perhaps the main reason why superresolution has not been attempted before on atmospheric sounders, is that these rely on the fact that the low resolution samples should be derived from a constant

underlying distribution (de Foy et al., 2009). When this is not the case, the smoothing introduced by oversampling is actually desirable. With the arrival of the wind-rotation technique, most of the variability observed for point source emitters can be corrected for, and therefore superresolution becomes viable for short-lived species as $NH_3$.

In Sec. 2 we introduce superresolution and demonstrate its effectiveness on measurements of the IASI sounder for a parameter related to (a constant) surface emissivity. Next we illustrate the application of what we coin 'wind-adjusted supersampling'

on an industrial point source of $NH_3$. In Sec. 3 we use ideas from McLinden et al. (2016) to provide a new type of $NH_3$ map,





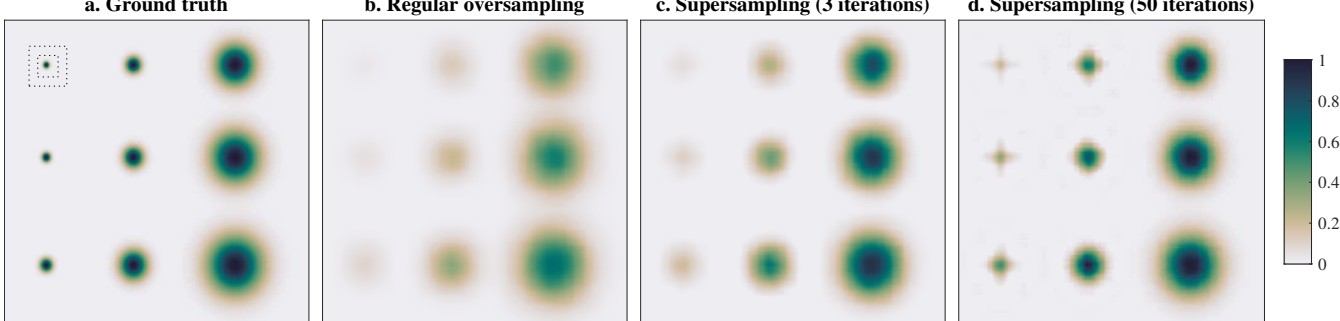

**Figure 1.** Illustration of the supersampling technique on synthetic data. The left panel depicts an imaginary ground truth, made up of two-dimensional Gaussian distributions, each with a different spread (0.5 to 40 pixels). The rectangles indicate the assumed footprint size of the measurements, varying between 7 and 13 pixels. Panel b shows the results of the common oversampling approach applied to 100000 measured scattered randomly over the area. Panels c and d provide the results of the supersampling technique after respectively 3 and 50 iterations.

one that is supersampled and wind-corrected at the same time. This map allows the identification of many new point sources in addition to the ones reported in Van Damme et al. (2018). We performed a detailed global analysis of this new map, which led to the identification and categorization of more than 500 point sources and which we present in Sec. 4.

## 2 Superresolved oversampling

The general superresolution problem does not have a unique solution, as the available low resolution measurements typically do not hold all the required information content (i.e. the problem is underdetermined). It is usually also overdetermined, because of measurement noise and, for our use case especially, because of temporal variability. As a consequence, there is no unique best algorithm, and a myriad of alternatives exist. For this study, we chose the Iterative Back-Projection algorithm (IBP, Irani and Peleg (1993)) as it takes a particular intuitive and simple form for single-pixel satellite observations, and allows addressing

the ill-determined nature of the problem. It proceeds as follows. Suppose we have a set of single pixel measurements $M_0$ of a spatially variable quantity. For the first iteration, the solution of the algorithm corresponds to the regular oversampling, which we will write as $SS_1 = OS_1 = \mathbf{OS}(M_0)$ ($SS_i$ stands for the solution of the supersampling obtained in iteration $i$ and $\mathbf{OS}$ stands for the oversampling operator). From this oversampled average, we then calculate simulated observations for each of the original individual observations, corresponding to what the instrument would see if the ground truth was $SS_1$. The entire set

of these simulated measurements will be denoted by $M_1 = \mathbf{M}(SS_1)$ (with $\mathbf{M}$ the operator that simulates the measurements). If the oversampled average $OS_1 = SS_1$ would correspond to the ground truth, then $M_1$ would clearly coincide with $M_0$. However, as oversampling typically smooths out the observations, this is generally not the case. An improved estimate of the average ($SS_2$), can be obtained by adding $\mathbf{OS}(M_0 - M_1)$ to the oversampled average, therefore correcting (partially) the




observed differences. This process then is repeated to obtain increasingly better estimates. The entire algorithm thus reads:

$$SS_1 = \mathbf{OS}(M_0) = OS_1 \qquad\qquad \rightarrow M_1 = \mathbf{M}(SS_1) \tag{1}$$

$$SS_2 = SS_1 + \mathbf{OS}(M_0 - M_1) = SS_1 + OS_1 - OS_2 \qquad\qquad \rightarrow M_2 = \mathbf{M}(SS_2) \tag{2}$$

$$\vdots$$

$$SS_k = SS_{k-1} + \mathbf{OS}(M_0 - M_{k-1}) = SS_{k-1} + OS_1 - OS_k \qquad\qquad \rightarrow M_k = \mathbf{M}(SS_k) \tag{3}$$

The solution converges to an average that is maximally consistent with the observations, i.e. $M_0 \approx M_k$ for sufficiently large $k$ (as shown in Elad and Feuer (1997), IBP converges to the maximum likelihood estimate whereby $M_0 - M_k$ is minimized). Figure 1 illustrates the algorithm on synthetic data with an idealized ground truth made up of 9 point sources (panel a), with a Gaussian spread between 0.5 and 40 pixels. The measurement footprint was assumed to be variable between 7 and 13 pixels.

The $SS_1$ (panel b), $SS_3$ (panel c) and $SS_{50}$ (panel d) averages illustrate well the convergence and strengths of the algorithm, which reproduces most of the point sources near-perfectly, and even partly resolves the smallest feature (compare also with Sun et al. (2018), Figure 8). Some small ringing effects are noticeable though after 50 iterations (best visible on a screen), which are the result of the undetermined nature of the problem (Dai et al., 2007).

An example on real data is shown in Figure 2, which shows part of the Sahara Desert and Mediterranean Sea. The quantity

on which the oversampling is applied, is the Brightness Temperature Difference (BTD) between the IASI channels at 1157 and 1168 cm$^{-1}$. This BTD, located in the atmospheric window, is sensitive to the sharp change in surface emissivity due to the presence of quartz (see Takashima and Masuda (1987), who also illustrate that the relevant feature is not seen in airborne dust). Being related to the surface, it can be assumed to be reasonably constant for each overpass of IASI (note that it is not entirely the case: sand dunes do undergo changes over time and surface emissivities can depend on the viewing angle and

can be affected by changes in moist content). Comparison with visible imagery (panel a) shows, as expected, that the largest BTD values ($> 4$ K) are associated with the most sandy areas. The other desert areas exhibit widely varying values, oceans are slightly negative. The oversampled average (panel b) captures most large features, down to about 5 km in size. Recalling that the footprint of IASI is a 12 km diameter circle at nadir, and elongates to an ellipse of up to 20 by 39 km at off-nadir angles, this example illustrates well why oversampling is such a powerful technique. However, the additional resolution brought by

the supersampling is clear, even after 3 iterations. The smallest features that can be distinguished are about 3–4 km (after 3 iterations, panel c) and 2–3 km (after 30 iterations, panel d) in diameter. That said, with increasing iterations, artifacts starts to appear due to enhancements of noise and the specific sampling of IASI (in particular, stripes parallel to the orbit track become apparent). Such overfitting to the data and a sensitivity to outliers, is often seen in maximum likelihood optimizations (Milanfar, 2010). It can therefore be advantageous to stop the algorithm after a few iterations (which can also be required for

computational reasons).



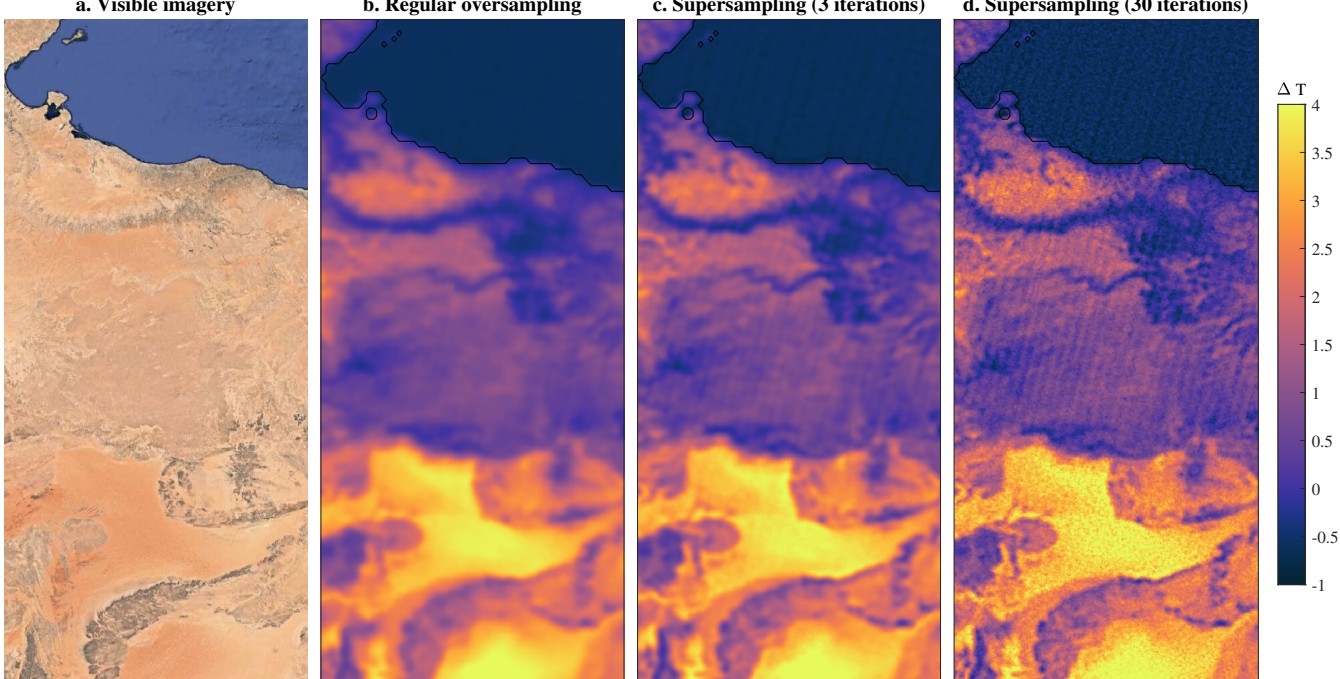

**Figure 2.** Illustration of the IBP superresolution technique on IASI observations of a BTD sensitive to surface quartz. All cloud-free observations of IASI for the period 2007-2018 were used for the averages. Panel a shows the corresponding visible imagery from Google Maps.

## 3 Wind-adjusted supersampling

In this section we illustrate the previously introduced supersampling on a wind-rotated $NH_3$ average centered around a point source. The ammonia plant at Horlivka (Gorlovka), Ukraine was chosen as a test case. This plant made the news in 2013 because of the major $NH_3$ leak that occurred on 6 August, killing five people and injuring many more. We refer to the Wikipedia article for a detailed description of the event, and a list of related newspaper articles (Wikipedia, 2019). The accident itself was not detected by IASI, but an abrupt drop in the average concentrations after the incident is seen in the satellite observations. In fact, after 2013 $NH_3$ enhancements are no longer detected at or near the plant. Fig. 3 illustrates the processes of oversampling, supersampling and wind-rotation on IASI data from 2007 to 2013. Each subpanel depicts the 120 km $\times$ 60 km area centered around the plant, from top to bottom:

**a. Gridded average** In the regular gridded average, each grid cell is assigned the arithmetic average of all observations whose center falls into the grid cell. This method only gives a faithful representation for larger grid cell sizes, whereas smaller grid sizes provide a higher resolution at the cost of larger noise. Here a grid size of $0.15° \times 0.15°$ was chosen. $NH_3$ enhancements are seen in wide area around the plant, with a maximum (north)east of the plant of $1 \cdot 10^{16}$ molec·cm$^{-2}$.



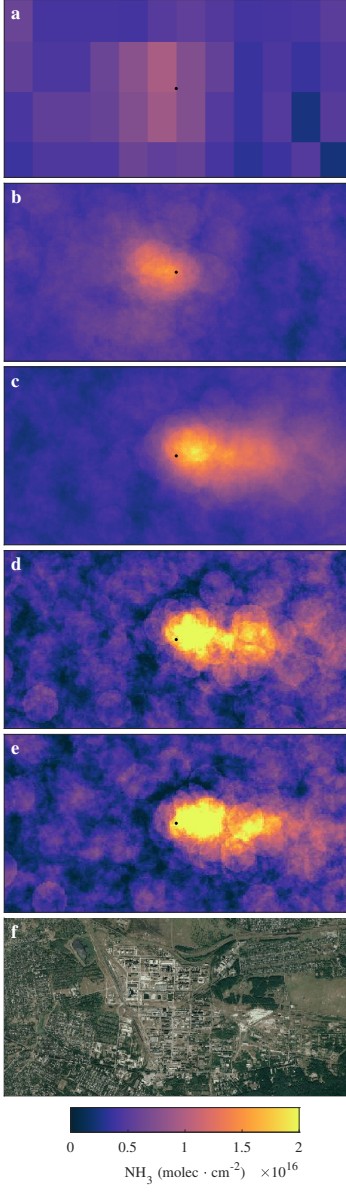

**Figure 3.** Averaging techniques illustrated on the ammonia plant at Horlivka on IASI NH$_3$ data between 2007 and 2013. From top to bottom: a. gridded average, b. oversampled average, c. wind-rotated oversampling, with the rotation center located at the maximum of (b), d. wind-rotated supersampling, with the rotation center located at the maximum of (b), e. wind-adjusted supersampling around the assumed source (center of the plant). f. Zoom in (factor 20) over the ammonia plant (data: Google Maps).

**b. Oversampled average**  Oversampling the daily maps before averaging increases the resolution and reveals the point source nature of the emission, with a maximum close to the plant (around $1.6 \cdot 10^{16}$ molec·cm$^{-2}$).



**c. Wind-rotated oversampling** The wind-rotation technique (Fioletov et al., 2015) consists of rotating the map of daily observations around a presumed point source, and along the direction of the wind direction at that point. The rotation was applied here to align the winds in the $x$-direction. Daily horizontal wind fields were taken from the ERA5 reanalysis (ERA5, 2019), and interpolated at an altitude equal to half of the boundary layer height. The $NH_3$ average shown in

this panel was obtained via oversampling applied to all the daily wind-rotated maps. It is important to note that such a distribution can no longer be interpreted as a geographical map, since each pixel is an average of measurements taken at different places. The only map element that is preserved is the distance to the point source. However, looking at the resulting distribution, the advantages brought by wind-rotation are obvious. Whereas in the normal oversampled average the $NH_3$ enhancements are scattered across, aligning the winds significantly enhances both the source and transport

(with a maximum of $2 \cdot 10^{16}$ molec·cm$^{-2}$).

**d. Wind-adjusted supersampling (i)** The figure in this panel was obtained from wind-rotated daily maps, as in the previous panel, but this time the average was calculated with 3 iterations of the IBP supersampling algorithm. As explained above, supersampling offers most benefits when the underlying distribution can be assumed reasonably constant, which is in part achieved by aligning the winds. The resolution is further increased, and as the plume is much less smoothed out,

maximum observed columns are also much higher ($3.3 \cdot 10^{16}$ molec·cm$^{-2}$).

**e. Wind-adjusted supersampling (ii)** In panels c and d, the point source location was taken from Van Damme et al. (2018), where the locations were determined based on the location of the maximums in the oversampled averages. In this last panel, the rotation was applied around the center of the presumed source (the chemical plant). The performance of the wind-rotation is further enhanced, yielding a distribution fully consistent with that of a single emitting point source whose emissions undergo transport in a fixed direction. The part of the plume located furthest from the source is a

20 bit off-axis, which is probably caused by inhomogeneities in the wind fields across the entire scene. This panel also illustrates the sensitivity of the rotation method to small shifts in the location of the center, a fact that we will exploit in the next section.

## 4 An $NH_3$ point source map

Having demonstrated the effectiveness of both the wind-rotation and supersampling approaches in revealing point sources, we are now in a position to introduce a new type of $NH_3$ map, specifically designed to track down point sources. It is based on a similar map presented in McLinden et al. (2016) for $SO_2$, but some important differences were introduced here to make it work for $NH_3$. The main idea of McLinden et al. (2016) is to treat each location on Earth as a potential point source and to assign it a value proportional to the downwind (the source) minus upwind (the background) column. In particular, for a given location,

a wind-rotated average is constructed first, similar to Fig. 3c. Representative average columns are then obtained downwind and upwind from the potential source (e.g. in boxes of $10 \times 10$ km$^2$). Finally, the difference of the up and downwind average is calculated, and this value is then used to represent the point source column at that specific location. While the method works





**Figure 4.** NH$_3$ point sources over Canada (top) and the US (bottom). The left panels show maps produced with a regular oversampled average, the right panels depict the corresponding NH$_3$ point source maps. The black circles indicate the identified point sources.





nicely for $SO_2$, this method proved to be only moderately successful when we applied it to the IASI $NH_3$ data. In particular, for those places where area sources dominate or where point sources are clustered over a too large area, local variation in the columns produce a noisy map, with many fictitious point sources.

We found that instead of the differences, the downwind average alone produced a more representative point source map. In addition, applying the method not on the oversampled average, but on the supersampled one, allows to increase the resolution. There are two key advantages offered by a point source map constructed in this way as opposed to a regular oversampled average: brighter point sources and smoother (lower) values over the background. The fact that point sources appear brighter is a direct consequence of the plume concentration achieved with wind-rotated supersampling, as shown in the previous section. Smoothing of the background is accomplished by the process of averaging the area downwind. However, by applying the method not on an oversampled average, but on a supersampled one, this smoothing is partially offset for point sources. The resulting point source map has a similar horizontal resolution as the oversampled map, but with increased averaged columns at the point sources and a smoother background distribution.

Examples over two selected regions in North America are shown in the right panels of Fig. 4. In these examples, the downwind averages were calculated in boxes extending from -5 to 5 km in the $y$-direction and 0 to 20 km in the $x$-direction. The left panels of the figure correspond to the regular oversampled averages. In panel a, which shows the oversampled average of the southern part of the Saskatchewan province of Canada, no point sources are apparent in the patchy $NH_3$ distribution. The corresponding point source map, shown in panel b, is smoother over areas dominated by the diffuse sources, where column variations are close to the measurement uncertainty. In addition, two bright spots are evident, which upon investigation coincide with the location of an ammonia plant (Belle Plaine) and a very large feedlot ($> 2$ km in length) near the town Lanigan. Looking back at the oversampled average, even with the advantage of hindsight, these sources can hardly be singled out. Panels c and d show the (south)western part of Kansas, US. It is an area well known for its cattle (Harrington and Lu, 2002). In Van Damme et al. (2018) several point sources associated with feedlots were isolated in Kansas and the rest of the High Plains region, but most of the area was found to be too diffuse to allow identification of individual point sources. The new $NH_3$ map facilitates greatly the attribution of these. This is due to the reduction in noise and the fact that the main point sources contrast much more with the background. An added benefit of this is that location of the maximums in the map is in general closer to the actual emission source than is the case in the oversampled map, making it easier to track down the suspected source with visible imagery, and therefore also to assign and identify the point source.

Displaced maximums that are seen in regular averages (wind-adjusted or not) can also be the result of transport, as noted by Van Damme et al. (2018), who found that especially for coastal sites, the shift can be as much as 20 km. The suspected reason is vertical uplift during transport, which makes $NH_3$ easier to detect and to measure (as can be seen in Fig. 3c) downwind of the source. The way the point source map is setup, corrects for the effects of transport as the columns are partially re-allocated back to their source by assigning the average downwind column to the point source. We have quantified the ability to locate sources on a careful selection of 36 industrial emitters. These were all chosen to be relatively isolated, with no nearby other industries or other sources, so that the actual emitting source is known with with confidence. In addition, only small to medium sized plants were chosen ($< 1$ km across), so that the precise location of the emission is known within a distance of about 500 m.





For the regular oversampled map, the sources were found within a median distance of 3.9 km and a mean of $5.4 \pm 3.7$ km. The furthest distance was 15.2 km. For the point source map, all but five sites were located within 3 km (with a median of 1.5 km, a mean of $2.1 \pm 1.7$ km and a maximum of 7.3 km), which confirms its improved performance.

A final advantage of the point source map is its performance in areas mildly affected by fires (e.g. in South East Asia,
Mexico, parts of South America). Certain hotspots due to fires, with a plume center of around 25 km, can look just like actual point sources. In the point source map, these often appear less bright and are blurred out over a wider area, with lower columns as compared to the oversampled average. On the other hand, as before, actual point sources appear brighter, and can emerge from the patchy $NH_3$ distribution that is characteristic for areas affected by fires. For that reason, comparing the oversampled and the point source map was found to be very useful for singling out point sources, especially in those areas with larger
background values. Example point sources are the ammonia plant in Campana (Argentina) and Bastos (Brazil), an important center of egg production. These were previously difficult to detect but are now easily identified.

## 5   Updated point source catalog

Using the methodology presented in the previous section, $NH_3$ point source maps of the world were constructed at a resolution of $0.01° \times 0.01°$ (land only). A few such maps were constructed varying the size of the averaging box, and the applied wind
speeds (either in the middle of the boundary layer or at 100 meter). While oversampling and backprojection are computationally not that demanding, we recall that the construction is based on the treating each gridcell of the $0.01° \times 0.01°$ map as potential point source, and therefore relies on the construction of wind-adjusted supersampled maps like Fig. 3e for each grid cell. Therefore, producing a worldmap at that resolution entails the generation of over 100,000,000 maps similar to Fig. 3e, each at a resolution of 1 km and each using more than 250,000 IASI measurements. A single point source map therefore takes more
than a month of computation. We decided to use all the available 2007–2017 $NH_3$ data both from IASI/Metop A (2007–2017) and IASI/Metop B (2013–2017), which helps to reduce the noise, even though it creates averages which are biased towards the last five years. The maps were then analyzed to provide an update of the point source catalog presented in Van Damme et al. (2018). We refer to it for a detailed description of the methodology for the identification and categorization of the point sources, as we used the same here. In brief, first, the global map is analyzed region per region, in search for $NH_3$ hotspots that are no
larger than 50 km across and that exhibit localized and concentrated enhancements compatible with a point source or dense cluster of point sources. Areas dominated by fires are excluded from the analysis. Analysis of areas with many sources or large ambient background concentrations, such as the Indo-Gangetic plain, is severely hampered, and reveals only the very large point sources. Isolated point sources in remote areas on the other hand can easily be picked up. The presence of a point source in the catalog should therefore not be seen as as a quantitative indicator of its emission strength. Note that in this study we did
not attempt to quantify the emission strengths of each. The categorization of the suspected point sources is performed using Google Earth imagery and third party information (mainly inventories of fertilizer plants and online resources). The original categories were: 'Agriculture', 'Fertilizer industry', 'Other industry', 'Natural' and 'Non-Determined'. Here we expanded the





**Figure 5.** Global distribution of NH₃ point sources and there categorization. The total number per cateogry is: Agriculture (215), Coking Industry (9), Explosive Industry (1), Fertilizer Industry (217), Geothermal Industry (3), Non-Determined Industry (21), Nickle Industry (11), Natural (1), Non-Determined (15), Urban (13)



number of categories considerably and in particular introduced an urban category and subdivided 'other industry' in five new categories as detailed below.

The new point source catalog is listed in Table A1 and illustrated on a world map in Fig. 5. Agricultural point sources were found to be invariably associated with CAFOs. Their number more than doubled, from 83 to 216, largely due to the increased attribution in areas of densely located point sources. For many of the previously tagged 'source regions', it was possible to resolve large individual emitters. This was the case in the US (particularly in the geographical region that corresponds to the High Plains Acquifer), Mexico and along the coast of Peru. Also notable are several newly exposed large feedlots in Canada and in eastern Australia. For the first time, agricultural point sources were also identified in China and Russia.

Industrial point sources, as before, are mainly associated with ammonia or urea-based fertilizer production (216, coming from 132) in Europe, North Africa and Asia. Industrial hotspots were categorized as fertilizer industry as soon as evidence was found of fertilizer production, even when there are clearly other industries present that may contribute. Separate categories were introduced for the previously identified soda ash, geothermal, nickle mining and coking industries, as additional examples were found for each. One ammonia plant in the US, associated with the manufacturing of explosives, was also assigned a separate category. Emission over unidentified industries were labeled 'Non-determined industry'.

An important new category is the 'Urban' one. Previously, localized emissions near Mexico City, Bamako (Mali) and Niamey (Niger) were noted. While these hotspots represent diffuse sources, they have been included in the catalog as the extent of the emissions in the relevant cities is sufficiently local, and sufficiently in excess of background values. Thanks to the improved source representation, clear enhancements were found in Kabul (Afghanistan) and 12 African urban agglomerations: Ouagadougou (Burkina Faso), Bamako (Mali), Kano (Nigeria), Niamey (Niger), Maiduguri (Nigeria), Khartoum-Omdurman (Sudan), Luanda (Angola), Kinshasa (Congo), Nairobi (Kenya), Addis Ababa (Ethiopia), Bamako (Mali) and Kampala (Uganda). Especially in Asia, atmospheric $NH_3$ is found in excess over most megacities, and with much larger columns than found in these African megacities. However, because of the much larger background columns, and much denser clusters of cities, these could not be singled out as was the case in Africa. Apart from industry, known urban sources of $NH_3$ include emissions from vehicles, human waste (waste treatment, sewers), biological waste (garbage containers) and domestic fires (including waste incineration) (Adon et al., 2016; Sun et al., 2017; Reche et al., 2015). At least the hotspot at Bamako is consistent with in situ measurements (Adon et al., 2016), that report year-round very high concentrations, between 28–73 ppb on a monthly averaged basis. Local conditions surely are key to explain why some cities in Africa exhibit much larger concentrations than others. Johannesburg (South Africa) for instance blends in completely in the background, with ambient values almost not larger than in the rest of South Africa. While outside of the scope of this paper, there is no doubt that the IASI $NH_3$ data could be further exploited to understand better the driving factors of urban emission on a global scale.

Other than at Lake Natron (Clarisse et al., 2019), no other natural $NH_3$ hotspots have been identified. For a number of presumed point sources no likely source could be attributed; however given their location (central US, Middle East, East Asia), these are most likely anthropogenic.





# 6   Conclusions

Oversampling is a technique now commonplace in the field of atmospheric sounding for achieving hyperresolved spatial averages, far beyond what the satellites natively offer. There is a class of algorithms referred to as superresolution that goes beyond oversampling, but these have until now only been applied to measurements of satellite imagers for surface parameters.

Here, we have shown that it is a viable method that can also be applied to the single pixel images taken by atmospheric sounders for short lived gases. We demonstrated this with measurements of a quartz emissivity feature over the Sahara Desert, for which a spatial resolution down to 2–3 km could be achieved.

Superresolution is a priori less suitable for measurements of atmospheric gases because of variations in their distribution, related to variations in transport. However, by aligning the winds around point source emitters, much of this variability can be

removed. In Sec. 3, we have shown the advantage of applying IBP superresolution on such wind-corrected data. The resulting averaged plumes originating from point sources not only reveal much more detail, maximum concentrations and gradients are also much larger, and presumably more realistic. Studies of atmospheric lifetime (e.g. Fioletov et al. (2015)), which rely on the precise shape of the dispersion, could potential benefit from this increase in accuracy.

Wind-adjusted superresolution images around point sources form the basis of the $NH_3$ point source map, which is an $NH_3$

average that simultaneously corrects for wind transport, accentuates point sources and smooths area sources. It was inspired by the $SO_2$ 'difference' map presented in McLinden et al. (2016), but as we do not look at differences, the $NH_3$ map still looks like an $NH_3$ total column distribution. However, other than for the identification of point sources, such a map is not easily exploitable, as it is a distorted representation of the reality that favors point sources. In depth analysis allowed us to perform a major update of the global catalog of point sources presented in Van Damme et al. (2018), with more than 500 point sources

identified and categorized. As a whole, this study further highlights the importance of point sources on local scales. The world map shows distinct patterns, with agricultural point sources completely dominant in America, in contrast to Europe and Asia where industrial point sources are prevalent. In Africa, $NH_3$ hotspots are mainly found near urban agglomerations.

While the point source catalog was established with a great deal of care, given its size, mistakes will inevitably be present, both in the localization of the point sources (due to e.g. noise in the data or $NH_3$ in transport) and in the categorization.

Improvements can probably best be achieved with feedback from the international community, with complementary knowledge on regional sources. For this reason, and to keep track of emerging emission sources, we have setup an interactive website, with the catalog and that allows to visualize the distribution and type of the point sources (http://www.ulb.ac.be/cpm/NH3-IASI.html). With the help of the community, we hope it can evolve as a go-to resource for information on global $NH_3$ point sources.

*Data availability.* The IASI $NH_3$ product is available from the Aeris data infrastructure (http://iasi.aeris-data.fr). It is also planned to be

operationally distributed by EUMETCast, under the auspices of the Eumetsat Atmospheric Monitoring Satellite Application Facility (AC-SAF ; http://ac-saf.eumetsat.int).





## Appendix A:  Point Source Catalog

**Table A1.** Updated point source catalog. The categories are abbreviated as: A = Agriculture, CI = Coking Industry, EI = Explosive Industry, FI = Fertilizer Industry, GI = Geothermal Industry, NDI= Non-Determined Industry, NI = Nickle Industry, SI = Soda ash Industry, N = Natural, ND = Non-Determined, U = Urban.

| Country | Lat | Lon | Name | Type | Country | Lat | Lon | Name | Type |
|---|---|---|---|---|---|---|---|---|---|
| Australia | -35.92 | 146.37 | Redlands | A | Australia | -34.66 | 146.49 | Merungle Hill | A |
| Australia | -34.46 | 147.77 | Springdale | A | Australia | -29.52 | 151.72 | Emmaville | A |
| Australia | -28.74 | 151.04 | Beebo | A | Australia | -27.46 | 151.13 | Grassdale | A |
| Australia | -27.15 | 151.54 | Moola | A | Australia | -26.90 | 149.84 | Morabi | A |
| Australia | -26.82 | 150.40 | Greenswamp | A | Belarus | 52.28 | 23.52 | Malyja Zvody | A |
| Belgium | 51.06 | 3.25 | Wingene | A | Bolivia | -17.34 | -66.28 | Cochabamba | A |
| Brazil | -21.93 | -50.77 | Bastos | A | Brazil | -19.79 | -44.69 | Carioca | A |
| Brazil | -4.98 | -42.76 | Teresina | A | Canada | 49.03 | -122.28 | Abbotsford | A |
| Canada | 49.86 | -112.87 | Picture Butte | A | Canada | 50.59 | -111.96 | Brooks | A |
| Canada | 50.90 | -113.37 | Stangmuir | A | Canada | 51.85 | -104.85 | Lanigan | A |
| Canada | 52.56 | -110.88 | Hughenden-Czar | A | Canada | 53.65 | -111.97 | Norma | A |
| Chile | -34.26 | -71.60 | Las Chacras | A | Chile | -33.95 | -71.64 | La Manga | A |
| China | 41.42 | 114.81 | Erdao Canal, Zhangbei (HE) | A | China | 41.84 | 115.83 | Xingtai Yong, Guyuan (HE) | A |
| China | 23.26 | 112.67 | Lianhuazhen (HI) | A | Dominican Republic | 19.43 | -70.54 | Moca - Tamboril | A |
| Emirates | 24.41 | 55.74 | Masaken | A | Emirates | 25.21 | 55.53 | Dubai | A |
| India | 11.29 | 78.14 | Namakkal | A | India | 16.75 | 81.65 | Tanaku | A |
| India | 17.39 | 78.61 | Hyderabad | A | Indonesia | -8.04 | 112.07 | Blitar City | A |
| Italy | 41.08 | 14.04 | Cancelo ed Arnone | A | Jordania | 32.13 | 36.27 | Dhlail Sub-District | A |
| Kazakhstan | 43.47 | 76.78 | North of Almaty | A | Malaysia | 5.21 | 100.48 | Sungai Jawi | A |
| Marocco | 33.87 | -6.88 | Temara | A | Mexico | 18.45 | -97.31 | Tehuacan | A |
| Mexico | 18.84 | -97.80 | Tochtepec | A | Mexico | 20.25 | -102.47 | Vista Hermosa de Negrete | A |
| Mexico | 20.68 | -99.93 | Ezequiel Montes | A | Mexico | 20.75 | -102.88 | Acatic | A |
| Mexico | 21.08 | -100.49 | San Antonio - La Canlea | A | Mexico | 21.21 | -102.41 | San Juan de Los Lagos | A |
| Mexico | 21.89 | -98.73 | Tampaon | A | Mexico | 22.03 | -102.30 | Aguascalientes | A |
| Mexico | 22.10 | -98.62 | Loma Alta | A | Mexico | 22.18 | -100.90 | San Luis Potosi | A |
| Mexico | 24.82 | -107.61 | Culiacancito | A | Mexico | 25.69 | -103.48 | Torreon | A |
| Mexico | 27.15 | -104.94 | Jiminez | A | Mexico | 27.39 | -109.89 | Obregon | A |
| Mexico | 28.20 | -105.43 | Delicias | A | Mexico | 32.46 | -116.80 | La Presa | A |
| Mexico | 32.51 | -115.22 | Puebla | A | Mexico | 32.61 | -115.63 | Santa Isabel | A |
| Peru | -16.54 | -71.89 | Vitor District | A | Peru | -16.42 | -72.28 | Majes | A |
| Peru | -13.46 | -76.09 | Alto Laran District | A | Peru | -12.97 | -76.43 | Quilmana District | A |
| Peru | -12.28 | -76.83 | Punta Hermosa | A | Peru | -11.94 | -77.07 | Carabayllo Disctrict | A |
| Peru | -11.53 | -77.23 | Huaral District | A | Peru | -11.30 | -77.42 | Irrigacion Santa Rosa | A |
| Peru | -11.05 | -77.56 | Tiroles | A | Peru | -8.15 | -78.97 | Trujillo | A |
| Peru | -7.99 | -79.20 | Chiquitoy | A | Peru | -7.25 | -79.48 | Guadalupe | A |
| Poland | 52.97 | 19.89 | Biezun | A | Russia | 50.78 | 35.87 | Rakitnoye | A |
| Russia | 51.12 | 41.51 | Novokhopyorsk | A | Russia | 54.67 | 61.35 | Klyuchi | A |
| Saudi Arabia | 24.10 | 48.92 | Haradh | A | Saudi Arabia | 24.19 | 47.45 | Al Qitar | A |
| Saudi Arabia | 24.22 | 47.93 | At Tawdihiyah | A | Saudi Arabia | 25.50 | 49.61 | Al Hofuf | A |
| South Africa | -26.62 | 28.28 | Ratanda | A | South Korea | 37.12 | 127.44 | Anseong - Icheon | A |
| Spain | 37.56 | -1.66 | Lorca - Puerto Lumbreras | A | Spain | 37.73 | -1.24 | Canovas | A |
| Spain | 38.40 | -4.87 | El Viso - Pozoblanco | A | Spain | 39.65 | -4.27 | Menasalbas | A |
| Spain | 41.12 | -4.21 | Mozoncillo | A | Spain | 41.95 | 2.21 | Vic - Manlleu | A |
| Spain | 40.87 | -0.03 | La Portellada | A | Spain | 41.93 | -1.21 | Tauste | A |
| Taiwan | 22.69 | 120.52 | Pingtung | A | Thailand | 13.30 | 101.26 | Nong Irun | A |
| Thailand | 13.46 | 99.70 | Thung Luang - Chom Bueng | A | Turkey | 37.26 | 33.29 | Alacati | A |
| Turkey | 37.57 | 34.02 | Eregli | A | Turkey | 37.78 | 32.53 | Konya | A |
| Turkey | 37.90 | 29.99 | Basmakci | A | Turkey | 38.73 | 30.57 | Afyonkarahisar | A |
| USA | 34.36 | -86.07 | Hopewell (AL) | A | USA | 32.68 | -114.08 | Wellton (AZ) | A |
| USA | 32.88 | -112.02 | Stanfield (AZ) | A | USA | 32.94 | -112.87 | Gila Bend (AZ) | A |
| USA | 33.33 | -111.70 | Higley (AZ) | A | USA | 33.37 | -112.70 | Palo Verde (AZ) | A |
| USA | 33.39 | -112.23 | Avondale (AZ) | A | USA | 33.17 | -115.59 | Calipatria (CA) | A |
| USA | 33.79 | -117.09 | San Jacinto (CA) | A | USA | 33.96 | -117.60 | Chino (CA) | A |
| USA | 35.23 | -119.09 | Bakersfield (CA) | A | USA | 36.08 | -119.43 | Tulare (CA) | A |
| USA | 36.29 | -120.28 | Coalinga - Huron (CA) | A | USA | 37.09 | -120.44 | Chowchilla (CA) | A |
| USA | 37.41 | -120.93 | Hilmar (CA) | A | USA | 38.24 | -122.73 | Petaluma (CA) | A |
| USA | 38.05 | -102.36 | Granada (CO) | A | USA | 38.07 | -103.76 | Rocky Ford (CO) | A |
| USA | 38.11 | -102.72 | Lamar (CO) | A | USA | 38.23 | -103.72 | Ordway (CO) | A |
| USA | 39.27 | -102.27 | Burlington (CO) | A | USA | 40.13 | -102.57 | Eckley - Yuma (CO) | A |
| USA | 40.21 | -103.78 | Fort Morgan (CO) | A | USA | 40.22 | -103.96 | Wiggins (CO) | A |
| USA | 40.36 | -104.53 | Greeley (CO) | A | USA | 40.55 | -103.30 | Atwood (CO) | A |



| Country | Lat | Lon | Name | Type | Country | Lat | Lon | Name | Type |
|---|---|---|---|---|---|---|---|---|---|
| USA | 40.78 | -102.94 | Iliff-Crook (CO) | A | USA | 32.34 | -83.94 | Montezuma (GA) | A |
| USA | 34.27 | -83.03 | Royston (GA) | A | USA | 43.13 | -96.29 | Sioux county (IA) | A |
| USA | 42.26 | -113.36 | Malta (ID) | A | USA | 42.33 | -114.05 | Oakley (ID) | A |
| USA | 42.75 | -114.65 | Jerome - Wendell (ID) | A | USA | 43.05 | -116.07 | Grand View (ID) | A |
| USA | 43.43 | -116.48 | Melba (ID) | A | USA | 43.66 | -112.11 | Roberts (ID) | A |
| USA | 43.83 | -116.90 | Parma (ID) | A | USA | 38.49 | -86.88 | Jasper (IN) | A |
| USA | 41.04 | -87.26 | Fair Oaks (IN) | A | USA | 37.03 | -100.87 | Liberal (KS) | A |
| USA | 37.24 | -100.91 | Seward county (KS) | A | USA | 37.44 | -101.32 | Ulysses (KS) | A |
| USA | 37.60 | -100.94 | Haskell county (KS) | A | USA | 37.85 | -100.87 | Garden City (KS) | A |
| USA | 37.91 | -100.40 | Cimarron (KS) | A | USA | 38.12 | -99.07 | Larned (KS) | A |
| USA | 38.30 | -100.89 | Scott county (KS) | A | USA | 38.39 | -98.80 | Great Bend (KS) | A |
| USA | 38.58 | -101.36 | Wichita county (KS) | A | USA | 38.60 | -100.47 | Shields (KS) | A |
| USA | 39.07 | -100.84 | Oakley (KS) | A | USA | 39.41 | -100.52 | Hoxie (KS) | A |
| USA | 39.76 | -97.76 | Scandia (KS) | A | USA | 39.85 | -98.32 | Burr Oak (KS) | A |
| USA | 40.48 | -93.39 | Lucerne (MO) | A | USA | 40.15 | -98.50 | Cowles (NE) | A |
| USA | 40.22 | -100.54 | McCook (NE) | A | USA | 40.57 | -99.52 | Westmark (NE) | A |
| USA | 40.62 | -98.90 | Newark (NE) | A | USA | 40.67 | -101.63 | Chase county (NE) | A |
| USA | 40.76 | -99.72 | Lexington (NE) | A | USA | 40.79 | -97.11 | Seward county (NE) | A |
| USA | 40.87 | -100.74 | Wellfleet (NE) | A | USA | 40.98 | -100.19 | Gothenburg (NE) | A |
| USA | 41.35 | -99.63 | Broken Bow (NE) | A | USA | 41.54 | -102.96 | Bridgeport (NE) | A |
| USA | 41.78 | -103.43 | Minatare (NE) | A | USA | 41.99 | -96.93 | Wisner (NE) | A |
| USA | 42.00 | -103.71 | Mitchell (NE) | A | USA | 42.01 | -98.15 | Elgin (NE) | A |
| USA | 42.43 | -96.86 | Allen (NE) | A | USA | 32.10 | -106.63 | Vado (NM) | A |
| USA | 32.57 | -107.27 | Hatch (NM) | A | USA | 32.92 | -103.23 | Lovington (NM) | A |
| USA | 33.28 | -104.44 | Dexter - Rosswell (NM) | A | USA | 34.51 | -106.78 | Veguita (NM) | A |
| USA | 39.08 | -119.26 | Lyon county (NV) | A | USA | 39.41 | -118.77 | Fallon (NV) | A |
| USA | 40.36 | -84.73 | Coldwater (OH) | A | USA | 36.56 | -102.20 | Griggs (OK) | A |
| USA | 36.64 | -101.36 | Guymon (OK) | A | USA | 36.70 | -101.08 | Adams (OK) | A |
| USA | 36.76 | -101.30 | Optima (OK) | A | USA | 36.88 | -101.60 | Hough (OK) | A |
| USA | 45.72 | -119.83 | Boardman (OR) | A | USA | 29.65 | -97.37 | Gonzales (TX) | A |
| USA | 32.07 | -98.39 | Dublin (TX) | A | USA | 33.14 | -95.35 | Hopkins county (TX) | A |
| USA | 34.01 | -102.37 | Amherst (TX) | A | USA | 34.09 | -102.00 | Hale Center (TX) | A |
| USA | 34.19 | -101.45 | Lockney (TX) | A | USA | 34.42 | -103.08 | Farwell (TX) | A |
| USA | 34.50 | -102.41 | Castro (TX) | A | USA | 34.63 | -101.86 | Happy - Tulia (TX) | A |
| USA | 34.75 | -102.46 | Hereford (TX) | A | USA | 35.02 | -102.36 | Deaf Smith (TX) | A |
| USA | 35.07 | -102.04 | Bushland (TX) | A | USA | 35.55 | -100.75 | Pampa (TX) | A |
| USA | 35.85 | -102.45 | Hartely (TX) | A | USA | 36.01 | -102.60 | Dalhart (TX) | A |
| USA | 36.03 | -102.08 | Cactus (TX) | A | USA | 36.05 | -102.28 | Dalhart (east) (TX) | A |
| USA | 36.16 | -101.60 | Morse (TX) | A | USA | 36.28 | -100.68 | Ochiltree (TX) | A |
| USA | 36.30 | -102.03 | Stratford (TX) | A | USA | 38.19 | -113.26 | Milford (UT) | A |
| USA | 39.38 | -112.60 | Delta (UT) | A | USA | 41.95 | -111.97 | Trenton (UT) | A |
| USA | 38.45 | -79.00 | Bridgewater (VA) | A | USA | 46.35 | -119.00 | Eltopia (WA) | A |
| USA | 46.37 | -120.07 | Yakima Valley - Sunnyside (WA) | A | USA | 46.52 | -118.94 | Mesa (WA) | A |
| USA | 47.01 | -119.09 | Warden (WA) | A | USA | 42.04 | -104.14 | Torrington (WY) | A |
| Venezuela | 10.05 | -68.09 | Tocuyito - Barrerita | A | Venezuela | 10.41 | -71.79 | La Concepcion | A |
| Vietnam | 10.46 | 106.42 | Tan An | A | Vietnam | 11.02 | 106.94 | Bien Hoa | A |
| Vietnam | 20.76 | 105.95 | Khoai Chau | A | China | 45.77 | 130.91 | Qitaihe (HL) | CI |
| China | 38.72 | 110.17 | Jinjiezhen (SN) | CI | China | 39.11 | 110.74 | Xinminzhen, Fugu (SN) | CI |
| China | 39.18 | 110.31 | Sunjiachazhen, Shenmu (SN) | CI | China | 39.27 | 111.07 | Shishanzecun, Fugu (SN) | CI |
| China | 35.90 | 111.44 | Xiangfen (SX) | CI | China | 37.08 | 111.79 | Xiaoyi (SX) | CI |
| Russia | 53.72 | 91.01 | Chernogorsky | CI | Russia | 54.30 | 86.15 | Bachatsky | CI |
| USA | 41.08 | -104.90 | Cheyenne (WY) | EI | Algeria | 35.83 | -0.32 | Arzew | FI |
| Algeria | 36.90 | 7.72 | Annaba | FI | Argentinia | -34.19 | -59.03 | Campana | FI |
| Bangladesh | 22.27 | 91.83 | Chittagong | FI | Bangladesh | 24.01 | 90.97 | Ashuganj | FI |
| Bangladesh | 24.68 | 89.85 | Tarakandi | FI | Belarus | 53.67 | 23.91 | Grodno | FI |
| Brazil | -25.53 | -49.40 | Curitiba | FI | Brazil | -10.79 | -37.18 | Laranjeiras | FI |
| Bulgaria | 42.02 | 25.66 | Dimtrovdgrad | FI | Bulgaria | 43.21 | 27.63 | Devnya | FI |
| Canada | 42.76 | -82.41 | Courtright | FI | Canada | 49.82 | -99.92 | Brandon | FI |
| Canada | 50.07 | -110.68 | Medicine Hat | FI | Canada | 50.44 | -105.22 | Belle Plaine | FI |
| Canada | 53.73 | -113.17 | Fort Saskatchewan | FI | China | 30.05 | 116.83 | Xiangyuzhen (AH) | FI |
| China | 30.50 | 117.02 | Anqing (AH) | FI | China | 30.88 | 117.74 | Tongling (AH) | FI |
| China | 32.43 | 118.44 | Lai'an (AH) | FI | China | 32.63 | 116.97 | Huainan (AH) | FI |
| China | 32.93 | 115.84 | Fuyang (AH) | FI | China | 33.06 | 115.30 | Linquan (AH) | FI |
| China | 24.54 | 117.64 | Longwen (FJ) | FI | China | 36.06 | 103.59 | Xigu - Lanzhou (GS) | FI |
| China | 38.38 | 102.07 | Jinchang (GS) | FI | China | 24.34 | 109.35 | Liuzhou (GX) | FI |
| China | 25.18 | 104.84 | Xingyi - Qianxinan (GZ) | FI | China | 26.61 | 107.48 | Fuquan (GZ) | FI |
| China | 27.17 | 106.74 | Xiaozhaibazhen (GZ) | FI | China | 27.29 | 105.34 | Yachizhen (GZ) | FI |
| China | 32.97 | 114.05 | Zhumadian (HA) | FI | China | 34.79 | 114.42 | Kaifeng (HA) | FI |
| China | 35.25 | 113.74 | Xinxiang (HA) | FI | China | 35.55 | 114.59 | Huaxian (HA) | FI |
| China | 30.34 | 111.64 | Zhijiang (HB) | FI | China | 30.43 | 115.25 | Xishui (HB) | FI |
| China | 30.45 | 111.49 | Xiaoting (HB) | FI | China | 30.50 | 112.88 | Qianjiang (HB) | FI |
| China | 30.78 | 111.82 | Dangyang (HB) | FI | China | 30.94 | 113.66 | Yingcheng - Yunmeng (HB) | FI |





| Country | Lat | Lon | Name | Type | Country | Lat | Lon | Name | Type |
|---|---|---|---|---|---|---|---|---|---|
| China | 31.22 | 112.29 | Shiqiaoyizhen (HB) | FI | China | 37.87 | 116.55 | Dongguang (HE) | FI |
| China | 38.13 | 114.74 | Shijiazhuang - Gaocheng (HE) | FI | China | 46.46 | 125.20 | Xinghuacun (Longfen) (HL) | FI |
| China | 46.75 | 129.54 | Haolianghe (HL) | FI | China | 47.17 | 123.63 | Hulan Ergi (HL) | FI |
| China | 27.59 | 111.45 | Heqing (HN) | FI | China | 27.71 | 112.54 | Xianxiang (HN) | FI |
| China | 29.40 | 113.11 | Yueyang (HN) | FI | China | 35.76 | 114.96 | Puyang (HN) | FI |
| China | 44.01 | 126.56 | Jilin (JL) | FI | China | 45.31 | 124.47 | Changshan (JL) | FI |
| China | 31.32 | 121.01 | Kunshan (JS) | FI | China | 31.43 | 119.83 | Yixing (JS) | FI |
| China | 31.98 | 120.51 | Zhangjiagang, Suzhou Shi, (JS) | FI | China | 32.22 | 118.77 | Dachang - Nanjing (JS) | FI |
| China | 34.36 | 118.31 | Xinyi (JS) | FI | China | 34.60 | 119.13 | Lianyungang (JS) | FI |
| China | 34.75 | 116.63 | Fengxian (JS) | FI | China | 40.76 | 120.83 | Huludao (Liaoning) | FI |
| China | 41.20 | 121.98 | Shuangtaizi, Panjin (LS) | FI | China | 38.07 | 108.98 | Nalin river (NM) | FI |
| China | 39.08 | 109.47 | Tuke Sumu (NM) | FI | China | 39.43 | 106.70 | Wuda - Hainan - Huinong (NM) | FI |
| China | 40.04 | 111.28 | Lamawanzhen (NM) | FI | China | 40.69 | 108.70 | Wulashan (NM) | FI |
| China | 40.70 | 111.50 | Hohhot (NM) | FI | China | 43.45 | 122.25 | Mulituzhen (NM) | FI |
| China | 47.94 | 122.83 | Zalatun (NM) | FI | China | 49.36 | 119.67 | Hulun Buir (NM) | FI |
| China | 38.46 | 106.07 | Yinchuan (NX) | FI | China | 38.89 | 106.42 | Shizuishan (NX) | FI |
| China | 36.48 | 101.49 | Huangzhong (QH) | FI | China | 36.75 | 95.25 | Chaerhan Salt Lake (QH) | FI |
| China | 28.75 | 105.38 | Naxi (SC) | FI | China | 30.00 | 103.83 | Dongpo (SC) | FI |
| China | 30.84 | 105.35 | Shehong (SC) | FI | China | 30.90 | 104.25 | Deyang - Guanghan - Xindu (SC) | FI |
| China | 34.91 | 118.48 | Linyi (SD) | FI | China | 35.00 | 117.24 | Mushizhen, Tengzhou (SD) | FI |
| China | 35.51 | 118.51 | Yinan (SD) | FI | China | 35.87 | 116.43 | Dongping (SD) | FI |
| China | 36.30 | 117.52 | Yanglizhen (SD) | FI | China | 36.35 | 116.15 | Liaocheng (SD) | FI |
| China | 36.90 | 117.43 | Shuizhaizhen (SD) | FI | China | 36.95 | 118.77 | Shouguang (SD) | FI |
| China | 37.09 | 119.03 | Houzhen (Shouguang) (SD) | FI | China | 37.16 | 116.38 | Pingyuan (SD) | FI |
| China | 37.46 | 116.22 | Decheng (SD) | FI | China | 34.28 | 108.53 | Xingping (SN) | FI |
| China | 34.41 | 109.77 | Guapozhen (SN) | FI | China | 35.10 | 110.72 | Xian de Linyi (SX) | FI |
| China | 35.45 | 112.60 | Beiliuzhen (SX) | FI | China | 35.66 | 112.84 | Zezhou - Gaoping (SX) | FI |
| China | 36.35 | 113.31 | Lucheng (SX) | FI | China | 36.37 | 112.87 | Tunliu (SX) | FI |
| China | 36.60 | 111.70 | Huozhou (SX) | FI | China | 37.27 | 113.62 | Pingsongxiang (SX) | FI |
| China | 37.55 | 112.18 | Jiaocheng (SX) | FI | China | 38.33 | 112.11 | Jingle (SX) | FI |
| China | 41.72 | 83.03 | Kuqa (XJ) | FI | China | 43.99 | 87.64 | Midong - Fukang (XJ) | FI |
| China | 44.40 | 84.95 | Kuytun (XJ) | FI | China | 44.88 | 89.21 | Wucaiwan (XJ) | FI |
| China | 23.73 | 103.21 | Kaiyuan (YN) | FI | China | 24.97 | 103.13 | Yiliang (YN) | FI |
| China | 25.76 | 103.86 | Huashan (YN) | FI | China | 30.23 | 120.64 | Xiaoshan, Hangzhou (ZJ) | FI |
| China | 38.26 | 114.40 | Lingshou (HE) | FI | China | 19.08 | 108.67 | Dongfang (HI) | FI |
| Colombia | 10.30 | -75.49 | Cartagena - Mamonal | FI | Croatia | 45.48 | 16.82 | Kutina | FI |
| Egypt | 29.66 | 32.32 | Ain Sukhna | FI | Egypt | 31.07 | 31.40 | Talkha | FI |
| Egypt | 31.26 | 30.09 | Abu Qir | FI | Emirates | 24.18 | 52.73 | Ruwais | FI |
| Georgia | 41.54 | 45.08 | Rustavi | FI | Germany | 51.86 | 12.64 | Piesteritz | FI |
| India | 8.72 | 78.14 | Tuticorin | FI | India | 12.92 | 74.84 | Mangalore | FI |
| India | 13.13 | 80.25 | Manali - Chennai | FI | India | 15.34 | 73.85 | Zuarinigar | FI |
| India | 16.96 | 82.00 | Bikkavolu - Balabhadrapuram | FI | India | 18.71 | 72.86 | Thal | FI |
| India | 19.03 | 72.88 | Trombai - Mumbai | FI | India | 20.32 | 86.64 | Paradip - Batighara | FI |
| India | 21.17 | 72.71 | Hazira - Surat | FI | India | 21.59 | 73.00 | Ankleshwar | FI |
| India | 22.39 | 73.10 | Vadodara | FI | India | 24.51 | 77.14 | Vijaipur | FI |
| India | 25.19 | 76.17 | Gadepan | FI | India | 25.56 | 82.05 | Phulphur | FI |
| India | 26.46 | 80.21 | Kanpur | FI | India | 27.23 | 95.33 | Namrup | FI |
| India | 27.84 | 79.91 | Shahjahanpur | FI | India | 28.24 | 79.21 | Aonla | FI |
| Indonesia | -7.16 | 112.64 | Gresik | FI | Indonesia | -6.39 | 107.43 | Derwolong - Cikampek | FI |
| Indonesia | -2.97 | 104.79 | Palembang | FI | Indonesia | 0.18 | 117.48 | Bontang City | FI |
| Indonesia | 5.23 | 97.05 | Lhokseumawe | FI | Iran | 27.56 | 52.55 | Asaluyeh | FI |
| Iran | 29.86 | 52.72 | Marvdasht | FI | Iran | 30.40 | 49.11 | Bandar Imam Khomeini | FI |
| Iran | 37.54 | 57.49 | Bojnourd | FI | Iraq | 30.18 | 47.84 | Khor Al Zubair | FI |
| Kazakhstan | 43.66 | 51.21 | Aktau | FI | Lituania | 55.08 | 24.34 | Jonava | FI |
| Lybia | 30.42 | 19.61 | Marsa el Brega | FI | Mexico | 17.99 | -94.54 | Cosolaecaque | FI |
| Mexico | 20.52 | -101.14 | Salamanca - Villagran | FI | Morocco | 33.10 | -8.61 | Jorf Lasfar | FI |
| Myanmar | 16.90 | 94.76 | Kangyidaunt | FI | Myanmar | 17.15 | 95.98 | Hmawbi | FI |
| Nigeria | 4.73 | 7.11 | Port Harcourt | FI | North Korea | 39.63 | 125.64 | Anju | FI |
| Oman | 22.64 | 59.41 | Sur Industrial Estate | FI | Pakistan | 24.81 | 67.24 | Bin Qasim | FI |
| Pakistan | 28.07 | 69.69 | Daharki | FI | Pakistan | 28.27 | 70.07 | Sadiqabad | FI |
| Poland | 50.30 | 18.23 | Kedzierzyn - Kozle | FI | Poland | 51.47 | 21.96 | Pulawy | FI |
| Poland | 53.58 | 14.55 | Police | FI | Qatar | 24.91 | 51.58 | Mesaieed | FI |
| Romania | 43.70 | 24.89 | Turnu Magurele | FI | Romania | 44.53 | 27.37 | Slobozia - Dragalina | FI |
| Romania | 46.52 | 24.49 | Targu Mures | FI | Romania | 46.52 | 26.94 | Bacau | FI |
| Romania | 46.84 | 26.51 | Savinesti - Roznov - Slobozia | FI | Russia | 44.67 | 41.91 | Nevinnomyssk | FI |
| Russia | 50.14 | 39.68 | Rossosh | FI | Russia | 51.93 | 47.89 | Balakovo | FI |
| Russia | 53.40 | 55.87 | Salavat | FI | Russia | 53.54 | 49.61 | Togliatti | FI |
| Russia | 54.08 | 38.18 | Novomoskovsk | FI | Russia | 54.96 | 33.33 | Dorogobuzh | FI |
| Russia | 55.36 | 85.96 | Kemerovo | FI | Russia | 57.88 | 56.17 | Perm | FI |
| Russia | 58.53 | 49.95 | Kirovo-Chepetsk | FI | Russia | 58.61 | 31.24 | Novgorod | FI |
| Russia | 59.15 | 37.80 | Cherepovets | FI | Russia | 59.40 | 56.73 | Berezniki | FI |
| Saudi Arabia | 27.08 | 49.57 | Al Jubayl | FI | Saudi Arabia | 29.32 | 35.00 | Haql | FI |




| Country | Lat | Lon | Name | Type | Country | Lat | Lon | Name | Type |
|---|---|---|---|---|---|---|---|---|---|
| Serbia | 44.87 | 20.60 | Pancevo | FI | Slovakia | 48.16 | 17.96 | Sala | FI |
| South Africa | -26.85 | 27.82 | Sasolburg | FI | South Africa | -26.57 | 29.16 | Secunda | FI |
| Spain | 37.19 | -6.91 | Huelva | FI | Spain | 38.67 | -4.06 | Puertollano | FI |
| Syria | 34.67 | 36.68 | Homs | FI | Trinidad and Tobago | 10.40 | -61.48 | Point Lisas | FI |
| Tunisia | 33.91 | 10.10 | Gabes | FI | Tunisia | 34.76 | 10.79 | Sfax | FI |
| Turkmenistan | 37.37 | 60.47 | Tejen | FI | Turkmenistan | 37.50 | 61.84 | Mary | FI |
| Ukraine | 46.62 | 31.00 | Odessa-Yuzhne | FI | Ukraine | 48.31 | 38.11 | Gorlovka | FI |
| Ukraine | 48.50 | 34.66 | Kamianske | FI | Ukraine | 48.94 | 38.47 | Severodonetsk | FI |
| Ukraine | 49.37 | 32.05 | Cherkasy | FI | Ukraine | 50.70 | 26.20 | Rivne | FI |
| USA | 34.82 | -87.95 | Cherokee (AL) | FI | USA | 42.41 | -90.57 | Massey (IO) | FI |
| USA | 36.37 | -97.79 | Etna (KS) | FI | USA | 30.09 | -90.96 | Donaldsonville (LA) | FI |
| USA | 47.35 | -101.83 | Beulah (ND) | FI | USA | 33.44 | -81.94 | Beech Island (SC) | FI |
| Uzbekistan | 40.10 | 65.30 | Navoi | FI | Uzbekistan | 40.46 | 71.83 | Ferghana | FI |
| Uzbekistan | 41.44 | 69.51 | Chirchik | FI | Venezuela | 10.07 | -64.86 | El Jose | FI |
| Venezuela | 10.50 | -68.20 | Moron | FI | Venezuela | 10.74 | -71.57 | Maracaibo | FI |
| Vietnam | 10.62 | 107.02 | Phu My | FI | Vietnam | 20.24 | 106.07 | Ninh Binh | FI |
| Italy | 42.87 | 11.62 | Mt. Amiata | GI | Italy | 43.22 | 10.91 | Larderello | GI |
| USA | 38.77 | -122.80 | The Geysers (CA) | GI | Tanzania | -2.49 | 36.06 | Lake Natron | N |
| China | 31.83 | 117.43 | Feidong (AH) | ND | China | 32.11 | 117.38 | Jianbei (AH) | ND |
| China | 32.39 | 117.61 | Gaotangxiang (AH) | ND | China | 29.91 | 115.34 | Fuchizhen (HB) | ND |
| China | 31.73 | 120.22 | Yuqizhen (JS) | ND | China | 37.53 | 105.71 | Zhongning (NX) | ND |
| China | 35.47 | 115.53 | Juancheng (SD) | ND | China | 33.00 | 106.97 | Hanzhong (SN) | ND |
| China | 34.88 | 111.17 | Pinglu (SX) | ND | China | 24.16 | 102.77 | Tonghai (YN) | ND |
| Spain | 41.63 | -4.71 | Valladolid | ND | Syria | 33.51 | 36.40 | East of Damascus | ND |
| Taiwan | 23.93 | 120.35 | Fangyuan | ND | USA | 37.19 | -86.73 | Morgantown (WV) | ND |
| Vietnam | 10.74 | 106.59 | Ho Chi Minh | ND | China | 36.00 | 103.28 | Yongjing (GS) | NDI |
| China | 26.55 | 104.88 | Zhongshan (GZ) | NDI | China | 33.42 | 113.62 | Wuyang (HA) | NDI |
| China | 46.19 | 129.36 | Dalianhezhen (HL) | NDI | China | 46.57 | 124.83 | Cheng'ercun, Ranghulu (HL) | NDI |
| China | 39.40 | 121.73 | Xiaochentun, Wafangdia (LN) | NDI | China | 41.83 | 123.93 | Fushun (LN) | NDI |
| China | 39.87 | 106.81 | Huanghecun (NM) | NDI | China | 40.64 | 109.69 | Baotou (NM) | NDI |
| China | 42.31 | 119.24 | Yuanbaoshanzhen (NM) | NDI | China | 37.88 | 106.15 | Wuzhong (NX) | NDI |
| China | 38.23 | 106.54 | Ningdongzhen (NX) | NDI | China | 35.64 | 110.95 | Hejin - Jishan - Xinjiang (SX) | NDI |
| China | 36.31 | 111.74 | Hongtong (SX) | NDI | Egypt | 29.94 | 32.47 | Al-Adabiya | NDI |
| India | 23.77 | 86.40 | Jharia | NDI | Iran | 35.40 | 53.16 | Nezami | NDI |
| Mauritania | 18.05 | -15.98 | Nouakchott | NDI | Mexico | 26.89 | -101.42 | Monclova | NDI |
| Russia | 51.44 | 45.90 | Saratov | NDI | South Africa | -26.05 | 29.36 | Springbok | NDI |
| Australia | -19.20 | 146.61 | Yabulu | NI | Brazil | -14.35 | -48.45 | Niquelandia | NI |
| Cuba | 20.64 | -74.89 | Moa | NI | Cuba | 20.67 | -75.57 | Nicaro | NI |
| China | 39.22 | 118.13 | Douyangu (HE) | SI | China | 37.32 | 97.33 | Delingha (QH) | SI |
| China | 29.46 | 103.84 | Wutongqiao (SC) | SI | Mexico | 25.78 | -100.56 | Garcia | SI |
| Poland | 52.75 | 18.17 | Janikowo | SI | Poland | 52.75 | 18.15 | Inowroclaw | SI |
| Romania | 44.99 | 24.28 | Stuparei | SI | Russia | 53.66 | 55.99 | Sterlitamak | SI |
| Turkey | 36.79 | 34.67 | Mersin | SI | Ukraine | 45.97 | 33.85 | Krasnoperekopsk | SI |
| USA | 35.67 | -117.35 | Searles Valley (CA) | SI | Afghanistan | 34.51 | 69.17 | Kabul | U |
| Angola | -8.82 | 13.32 | Luanda | U | Burkina Faso | 12.35 | -1.58 | Ouagadougou | U |
| Congo | -4.39 | 15.32 | Kinshasa | U | Ethiopia | 9.02 | 38.71 | Addis Ababa | U |
| Kenya | -1.27 | 36.87 | Nairobi | U | Mali | 12.59 | -7.99 | Bamako | U |
| Mexico | 19.45 | -99.07 | Mexico City | U | Niger | 13.55 | 2.12 | Niamey | U |
| Nigeria | 11.88 | 13.17 | Maiduguri | U | Nigeria | 12.03 | 8.50 | Kano | U |
| Sudan | 15.65 | 32.55 | Omdurman - Khartoum | U | Uganda | 0.30 | 32.55 | Kampala | U |

*Author contributions.* L.C. conceptualized the study, wrote the code, prepared the figures and drafted the manuscript. L.C. and M.V.D. updated the point source catalog. All authors contributed to the text and interpretation of the results.

*Competing interests.* No competing interests are present.





*Acknowledgements.* IASI is a joint mission of EUMETSAT and the Centre National d'Études spatiales (CNES, France). It is flown on board the Metop satellites as part of the EUMETSAT Polar System. The IASI L1c and L2 data are received through the EUMETCast near-real-time data distribution service. L.C. is a research associate supported by the Belgian F.R.S-FNRS. The research was also funded by the Belgian State Federal Office for Scientific, Technical and Cultural Affairs (Prodex arrangement IASI.FLOW). The IASI $NH_3$ product is available

5   from the Aeris data infrastructure (http://iasi.aeris-data.fr).



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
