# Peer review of "Tracking down global $NH_3$ point sources with wind-adjusted superresolution"

_Atmospheric Measurement Techniques, 2019_

## Referee Comment (RC1) · Anonymous Referee #1 · 18 Jun 2019

This paper describes a new method to obtain high resolution NH3 columns from satellite by combining oversampling with wind-rotation. A global inventory of point sources is then developed by considering changes in columns for upwind and downwind locations. The science seems sound, the method is novel and the product will be very useful to researchers. Furthermore, the paper was very well written and a pleasure to read. I am happy to recommend it for publication.

In terms of references, I would suggest adding the following 2 citations: Page 3: The following is one of the early applications of oversampling: Russell, Ashley R., Lukas C. Valin, Eric J. Bucsela, Mark O. Wenig, and Ronald C. Cohen. "Space-based Constraints on Spatial and Temporal Patterns of NO x Emissions in California, 2005−2008." Environmental science & technology44, no. 9 (2010): 3608-3615.

[Figure]

Page 8, Line 1: As far as I am aware, the following introduced rotation first: Valin, L. C., A. R. Russell, and R. C. Cohen. "Variations of OH radical in an urban plume inferred from NO2 column measurements." Geophysical Research Letters 40, no. 9 (2013): 1856-1860.

My only other comments are minor proof-reading comments. Page 1, Line 1: is one *of* the primary, or is *a* primary Page 1, Line 6: do not use "allows to . . ." in English (see also line 15 and page 10, line 5). Allows needs a subject (allows you to) – which you probably don't want here. Maybe "enables the"? I'm not much of a fan of "so-called" – but that may be just personal preference. Page 1, Line 20: what about: "diffuse background with higher concentrations" or something like that?

Fig 1 Caption: "measurements" not "measured" Page 10, Line 19: town *of* Lanigan Page 10, Line 25: maxima not maximums, *the* location Page 10, Line 34: fix "with with" Page 11, Line 14: *by* varying Page 14, Line 1: This sentence seemed a bit hyperbolic Page 14, Line 27: "and that allows" needs fixing. "go-to" sounds a bit clumsy.

[Figure]

---

## Referee Comment (RC2) · Anonymous Referee #2 · 18 Jun 2019

This is a very-well written, and innovative study on how to use statistical techniques on long-time series to identify point sources of NH3 emissions across the globe. The paper is an extension of an earlier study published in Nature (Van Damme et al., 2018), and shows that adding information on winds derived from ECMWF's ERA re-analysis, allows to further increase the statistical power to discriminate point sources from the background signal.

The publication is build up in a logical well-chosen manner, examples (even if not pertaining per se to NH3) are well chosen. Somewhat surprising is that in this paper the authors do not estimate source strengths and uncertainties related to the point sources, for reasons not entirely clear to me. In contrast, the earlier Van Damme (2018) did provide such estimates (+uncertainties) so I do not see a strong reason why this paper

wouldn't- of course provided that everything works well.

In this context my main concerns are following:

- Mass conservation. Figure 3 nicely shows how oversampling and supersampling show enhanced plumes strength (as expressed by the maximum values). Given the short life-time of $NH_3$ (likely short due to the abundant presence of sulfate aerosol), one can assume that the average column values in the 60x120 km domain are mostly (entirely?) determined by the local source. Can the authors demonstrate that the domain average (or integrated) $NH_3$ columns are conservative across cases a) through e). Have such screening been performed for all identified large sources, and what was the result? With other words can we be sure that the algorithm does not artificially add mass, and be used to receive source strengths?

- This publication is an extension of the previous paper by Van Damme, which makes an important statement on the possible underestimation in inventories like EDGAR of nearly all agricultural and industrial point sources. As this paper is adding even more source, it would imply that the problem could be even aggravated. However, in none of these 2 papers an analysis is made of the potential impacts on regional and global emission budgets. I can easily imagine that the spatial allocation data used in inventories are not realistically representing a 0.1x0.1 degree resolution, but that 'point source' emissions are smeared out over larger areas. While the lack of spatial information in itself a serious problem, it may be less an issue for larger scale model analysis. It would be extremely helpful if the current paper could 1) provide quantitative information on derived emission strengths, similar to the previous paper 2) provide regional/global statistics of the aggregated amounts of annual point source emissions versus those in EDGAR and compared to all emissions, to get a better impression on how these new data would change our view on the global $NH_3$ budget.

I recommend publication of this paper, after taken into account my concerns.

Minor comments:

P2 l. 14 what were these adverse effects?

P2 l. 15 It is also related to other pollutant becoming relatively less important.

P2 l. 24 Clarify what is meant with conservative residence time. Van Damme varied between 1, 12 and 48 hours. I presume you meant 48 hours- as this would imply the lowest emission rate? Not for this paper, but you could get a better handle on the lifetime issue by collaborating with one or more modellers and relate lifetime to column and emission rates.

P3 l. 16 'reduces spread and contribution of nearby source': I didn't get it. Explain better.

P 3 l. 30 What is meant with a constant underlying distribution? Of what? I didn't get it.

p. 5 l. 30 I haven't seen what is the case for NH3, only few iteration or many? And why?

p.6 l. 9 As described above the example seems to add 'mass' to with the oversampling/super sampling. The authors should show whether this is the case or not.

p. 6 l. 31 If understand it well this is discussing the McLInden approach (but not yours). 100 km2 is quite a large area to calculate background and signal of point sources.

p. 8 l. 25 what is meant with an NH3 map. Concentration/column or emission?

P 10 l. 3. Noisy map and fictitious sources. How do you know that? Are you still speaking about 10x10 km areas for which oversampling/supersampling would create a noisy map?

p. 10 l. 4 It sounds counterintuitive that only looking at downwind concentrations an improved point source map can be improved. What would this mean for the retrieved emission values? Some more theoretical foundation for this approach would be valuable.

p. 10 l. 23 'The new NH3 map'. It would help the reader if you could give a better name to this map, describing what it really is. Something like 'satellite derived source attribution map'- it should be made clear that this is a calculated map- not something that is directly observable by the satellite instrument.

p. 11 l. 3 improved performance in geo-allocation of the sources.

p. 11 l. 4 point source map? See earlier comment. Use unique name for this product. I think it is more than a point source map (in the sense that there is quantitative information on source strength).

p. 11 l. 14 0.01x0.01 degree corresponds roughly to 1-1 to 2-2 km? Maybe helpful to give the reader a feeling for this.

p. 11 l. 16 I am wondering if there is not something smarter possible, based on a pre-screening of all available IASI observations. If no elevated concentrations are found in any data point it is not likely to be a relevant points source. Possibly for discussion or future work. Or maybe I understood it wrong, and you are describing what you don't want to do?

p. 11 l 23 what is meant with a single point source map? A single year? A single source? Clarify.

p. 11 l.29/30 This is confusing as statements are made on disagreement with emission inventories.

P 13 l. 26. What would be the equivalent retrieved concentration (with some reasonable assumption on BL height).

---

## Author Comment (AC1) · 17 Jul 2019

This paper describes a new method to obtain high resolution NH3 columns from satellite by combining oversampling with wind-rotation. A global inventory of point sources is then developed by considering changes in columns for upwind and downwind locations. The science seems sound, the method is novel and the product will be very useful to researchers. Furthermore, the paper was very well written and a pleasure to read. I am happy to recommend it for publication.

*Thank you very much for your review, your appreciation and positive assessment of the paper. Thank you also for the two references and corrections to the English – we have followed all suggested changes.*

In terms of references, I would suggest adding the following 2 citations:

Page 3: The following is one of the early applications of oversampling: Russell, Ashley R., Lukas C. Valin, Eric J. Bucsela, Mark O. Wenig, and Ronald C. Cohen. "Space-based Constraints on Spatial and Temporal Patterns of NO x Emissions in California, 2005–2008." Environmental science & technology 44, no. 9 (2010): 3608-3615.

Page 8, Line 1: As far as I am aware, the following introduced rotation first: Valin, L. C., A. R. Russell, and R. C. Cohen. "Variations of OH radical in an urban plume inferred from NO2 column measurements." Geophysical Research Letters 40, no. 9 (2013): 1856-1860.

*We have added these references now.*

My only other comments are minor proof-reading comments.

Page 1, Line 1: is one *of* the primary, or is *a* primary

*Corrected*

Page 1, Line 6: do not use "allows to . . ." in English (see also line 15 and page 10, line 5). Allows needs a subject (allows you to) – which you probably don't want here. Maybe "enables the"?

*After having researched a bit the alternatives, in the end we went for the active form "Oversampling allows one to increase the spatial resolution of averaged satellite data, beyond what the satellites natively offer." (so we added a subject) instead of the passive voice "Oversampling enables the spatial resolution of averaged satellite data to be increased" . We have also went through the rest of the text and fixed similar occurrences.*

I'm not much of a fan of "so-called" – but that may be just personal preference.

*We removed this from the abstract, and also once in the text.*

Page 1, Line 20: what about: "diffuse background with higher concentrations" or something like that?

*Corrected*

Fig 1 Caption: "measurements" not "measured"

*Corrected*

Page 10, Line 19: town *of* Lanigan

Corrected

Page 10, Line 25: maxima not maximums, *the* location

Corrected (here and in two other places)

Page 10, Line 34: fix "with with"

Corrected

Page 11, Line 14: *by* varying

Corrected

Page 14, Line 1: This sentence seemed a bit hyperbolic

Yes, we agree and changed "*far beyond*" into "*beyond*".

Page 14, Line 27: "and that allows" needs fixing. "go-to" sounds a bit clumsy.

We have reformulated these last two sentences now as follows: *For this reason, and to keep track of emerging emission sources, we have setup a website, with an interactive global map, visualizing the distribution, type and time evolution of the different point sources (http://www.ulb.ac.be/cpm/NH3-IASI.html). With the help of the community, we hope it can become a useful resource for information on global $NH_3$ point sources.*

---

## Author Comment (AC2) · 17 Jul 2019

This is a very-well written, and innovative study on how to use statistical techniques on long-time series to identify point sources of NH3 emissions across the globe. The paper is an extension of an earlier study published in Nature (Van Damme et al., 2018), and shows that adding information on winds derived from ECMWF's ERA re-analysis, allows to further increase the statistical power to discriminate point sources from the background signal. The publication is build up in a logical well-chosen manner, examples (even if not pertaining per se to NH3) are well chosen. Somewhat surprising is that in this paper the authors do not estimate source strengths and uncertainties related to the point sources, for reasons not entirely clear to me. In contrast, the earlier Van Damme (2018) did provide such estimates (+uncertainties) so I do not see a strong reason why this paper wouldn't- of course provided that everything works well.

*Thank you very much for your positive assessment of the paper and the detailed review. We have addressed all comments below, and where possible updated the text (this includes addressing the comment on mass conservation). The main exception is the comment on providing emission estimates. As we argue below, we think that such an analysis is largely beyond the scope of this paper, whose focus is on the new detection methodology.*

In this context my main concerns are following:

- Mass conservation. Figure 3 nicely shows how oversampling and supersampling show enhanced plumes strength (as expressed by the maximum values). Given the short lifetime of NH3 (likely short due to the abundant presence of sulfate aerosol), one can assume that the average column values in the 60x120 km domain are mostly (entirely?) determined by the local source. Can the authors demonstrate that the domain average (or integrated) NH3 columns are conservative across cases a) through e). Have such screening been performed for all identified large sources, and what was the result? With other words can we be sure that the algorithm does not artificially add mass, and be used to receive source strengths?

*Thank you very much for bringing up this question, which indeed should be addressed in this paper. The answer is not so simple however, as it depends whether it is discussed with respect to the spatial grid (ground truth) or with respect to the measurements. In fact, none of the procedures is mass conserving with respect to the ground truth, due to the finite number of measurements and their coarse resolution. In practice however, mass can be assumed to be conserved. In addition, in measurement space, supersampling is strictly mass conserving in the limit of a large number of iterations (we verified this on the examples of Fig 1, 2, and 3, as a way of verifying that the computer code did not contained any bugs). We have now added the values of the average columns in each subpanel in Figure 3, by means of illustration, and discussed the conservation issue in some length at the end of section 3:*

*One useful property of the different procedures is that they all approximately conserve the quantity that is being averaged, i.e. the averaged quantity in each grid is approximately the same as the average quantity in the grid representing the ground truth, given sufficient number of measurements across the entire grid. When the number of measurements is low, this can break down dramatically, as can be seen with the extreme example of single high-value measurement over an isolated point source. When a gridded average is made from this single measurement onto a coarse grid (e.g. 5° × 5°), the entire grid cell containing the measurement will be associated with this high value, thus yielding an overestimation of the reality. A strict conservation is therefore not possible in general, as not enough information is contained in the original measurements to reconstruct the ground truth perfectly, even on average. That being said, supersampling conserves quantity with respect to the original*

*measurements, when the number of iterations is large enough. This is a consequence of the fact that the backprojected measurements converge to the actual measurements, and therefore also their averages. Finally note that wind-rotation does not alter quantity in anyway, as rotation simply redistributes the measurements to different locations on the grid. The average total NH₃ columns are indicated on each subpanel of Fig 3. The average of the ungridded measurements within the considered box equals 5.23 x 10¹⁵ molec·cm⁻². As can be seen the largest change in average column is caused by the rotation procedure, but this is simply an artifact caused by limiting the average to a square box around a point source (instead of a circle). This example illustrates that in practice, with differences smaller than one percent, the different gridding procedures can be assumed to conserve quantity.*

- This publication is an extension of the previous paper by Van Damme, which makes an important statement on the possible underestimation in inventories like EDGAR of nearly all agricultural and industrial point sources. As this paper is adding even more source, it would imply that the problem could be even aggravated. However, in none of these 2 papers an analysis is made of the potential impacts on regional and global emission budgets. I can easily imagine that the spatial allocation data used in inventories are not realistically representing a 0.1x0.1 degree resolution, but that 'point source' emissions are smeared out over larger areas. While the lack of spatial information in itself a serious problem, it may be less an issue for larger scale model analysis. It would be extremely helpful if the current paper could 1) provide quantitative information on derived emission strengths, similar to the previous paper 2) provide regional/global statistics of the aggregated amounts of annual point source emissions versus those in EDGAR and compared to all emissions, to get a better impression on how these new data would change our view on the global NH3 budget. I recommend publication of this paper, after taken into account my concerns.

The focus of this paper is on introducing new methodologies for averaging satellite data and for the identification of point sources. In the second part of the paper the strengths of the new approach are demonstrated on NH₃, and we show that with the new method, we can identify twice the amount of NH₃ point sources compared to regular oversampling. The paper was written specifically with AMT in mind, as its focus is on introducing this new methodology. As such, it already represents a significant body of research, code writing, data analysis and computational effort. In our opinion, the topic and results constitute a well-separated entity that deserves to be published separate from a quantitative derivation, analysis and discussion of emissions. While we agree that what the reviewer asks is important, and can in principle be done, it is not something that we wanted to do in this paper. In addition, doing so would entirely draw away the attention from the focus of the paper, which is on introducing a new detection method for point sources.

Minor comments:

P2 l. 14 what were these adverse effects?

The adverse effects of decreasing $NO_x$ and $SO_2$ emissions, is that these have been shown to increase NH3 emissions and/or concentrations. We have clarified the sentence.

P2 l. 15 It is also related to other pollutant becoming relatively less important.

We do not understand how this comment relates to l15 as here we state that the regulative framework of NH3 is limited due to the historical relative difficulty in measuring $NH_3$.

P2 l. 24 Clarify what is meant with conservative residence time. Van Damme varied between 1, 12 and 48 hours. I presume you meant 48 hours- as this would imply the lowest emission rate? Not for this

paper, but you could get a better handle on the lifetime issue by collaborating with one or more modellers and relate lifetime to column and emission rates.

*We meant 12 hours, which is already above the mean value reported (and thus indeed implies that our corresponding emission estimate is a lower bound). We have clarified the sentence by adding this 12 hour value.*

P3 l. 16 'reduces spread and contribution of nearby source': I didn't get it. Explain better.

*The entire sentence reads: "As we will also demonstrate, this reduces the overall spread of the transported pollutants and reduces contributions of nearby sources." At this point of the introduction, this is just anticipating what is about to come. The point is developed in detail in section 3, and so now we explicitly refer to it.*

P 3 l. 30 What is meant with a constant underlying distribution? Of what? I didn't get it.

*This sentence means that superresolution is only viable when each low resolution image is derived from the same reality, i.e. that the underlying distribution does not change in time. We have replaced "constant" with "an underlying distribution that does not change in time".*

p. 5 l. 30 I haven't seen what is the case for NH3, only few iteration or many? And why?

*Too many iterations result in overfitting on the data, especially for $NH_3$, which has a large measurement uncertainty. In practice, we found that three iterations is a good compromise between smoothness and increasing the resolution. This was already partially covered in section 3, point d, but we have now added this sentence: "Note that in general for $NH_3$, 3 iterations of the IBP algorithm seems to offer a good compromise between increasing the resolution of the average, without introducing artefacts related to overfitting."*

p.6 l. 9 As described above the example seems to add 'mass' to with the oversampling/super sampling. The authors should show whether this is the case or not.

*At first sight, it might seem that oversampling or supersampling adds mass in this example. This is a visual effect: while the mass clearly increases inside the $NH_3$ plume, it also seen to decrease outside the plume, in a much larger area. In response to the very first comment of the review, we have added the average column on each subpanel, and added a discussion on mass conservation (see above).*

p. 6 l. 31 If understand it well this is discussing the McLinden approach (but not yours). 100 km2 is quite a large area to calculate background and signal of point sources.

*This is correct. 10 x 10 $km^2$ is large, but definitely not too large, given the lifetime of $SO_2/NH_3$.*

p. 8 l. 25 what is meant with an NH3 map. Concentration/column or emission?

*The map, being built from averages of downwind maps of columns, is also a column map; as further explained in that section (see also the example provided in Fig 4).*

P 10 l. 3. Noisy map and fictitious sources. How do you know that? Are you still speaking about 10x10 km areas for which oversampling/supersampling would create a noisy map?

It is not the oversampling/supersampling as such that creates a noisy map, but the application of the McLinden et al. approach of calculating differences up and downwind. As explained in the text, the problem comes from larger area sources, which produce a slowly varying NH3 distribution. Small local differences are amplified in the McLinden et al. approach, which relies on differences between neighbouring column averages.

p. 10 l. 4 It sounds counterintuitive that only looking at downwind concentrations an improved point source map can be improved. What would this mean for the retrieved emission values? Some more theoretical foundation for this approach would be valuable.

From the discussion around figure 3, it can already be intuitively understood that looking at the downwind plume only, the map as described will show large local enhancements around point sources. The map has nothing to do with emission values. The only values that represent a reality are the values just above the point sources, representing the mean value of the downwind plume. As also written in the conclusion: "*However, other than for the identification of point sources, such a map is not easily exploitable, as it is a distorted representation of the reality that favours point sources.*"

p. 10 l. 23 'The new NH3 map'. It would help the reader if you could give a better name to this map, describing what it really is. Something like 'satellite derived source attribution map'- it should be made clear that this is a calculated map- not something that is directly observable by the satellite instrument.

In fact, throughout the text we have consistently referred to the map as an "NH3 point source map", which we believe is a term that covers quite well the meaning of the map. We have now also used this terminology in this sentence.

p. 11 l. 3 improved performance in geo-allocation of the sources.

We have added "in geo-allocation of the sources" in this sentence.

p. 11 l. 4 point source map? See earlier comment. Use unique name for this product. I think it is more than a point source map (in the sense that there is quantitative information on source strength).

As explained above, we believe that this term is appropriate. It does give some quantitative information just above the point sources, but that quantitative aspect is also not reflected in the term 'satellite derived source attribution map' (and in fact the latter term does not express the fact that the map specifically is designed to highlight the point sources).

p. 11 l. 14 0.01x0.01 degree corresponds roughly to 1-1 to 2-2 km? Maybe helpful to give the reader a feeling for this.

Yes that is correct – we have added this info now: "corresponding to a horizontal resolution of the order of 1-2 km".

p. 11 l. 16 I am wondering if there is not something smarter possible, based on a prescreening of all available IASI observations. If no elevated concentrations are found in any data point it is not likely to be a relevant points source. Possibly for discussion or future work. Or maybe I understood it wrong, and you are describing what you don't want to do?

You did understand this correctly. Yes, it is definitely possible to be more selective by prescreening, and removing entire regions, but the danger always exist to miss weak point sources (recall that we can detect even very weak point sources, as long as they increase NH3 locally).

p. 11 l 23 what is meant with a single point source map? A single year? A single source? Clarify.

A single point source map is a map like in Figure 4. For this study, as explained in the beginning of that section, we constructed several ones: "*A few such maps were constructed varying the size of the averaging box, and the applied wind speeds (either in the middle of the boundary layer or at 100 meter).*"

p. 11 l.29/30 This is confusing as statements are made on disagreement with emission inventories.

In this paper we strictly deal with a qualitative detection of point sources, and make no statements on disagreement with emission inventories (unlike in Van Damme et al. , 2018). This sentence reinforces this, stating that the presence of a point source in the catalog should not be seen as a quantitative indicator of its emission strength.

P 13 l. 26. What would be the equivalent retrieved concentration (with some reasonable assumption on BL height).

We assume the reviewer means equivalent column. Assuming an approximate conversion (from a standard model of) $3 \times 10^{15}$ molec/cm$^2$ per ppb this would be of the order of 1 to 2 $\times 10^{17}$ molec/cm$^2$.